# A New Era for PET/CT: Applications in Non-Tumorous Renal Pathologies

**DOI:** 10.3390/jcm13164632

**Published:** 2024-08-07

**Authors:** Serin Moghrabi, Ahmed Saad Abdlkadir, Nabeela Al-Hajaj, Gopinath Gnanasegaran, Rakesh Kumar, Ghulam Syed, Murat Fani Bozkurt, Saad Shukri, Shahed Obeidat, Aysar Khalaf, Mohammed Shahait, Khalsa Al-Nabhani, Akram Al-Ibraheem

**Affiliations:** 1Department of Nuclear Medicine and PET/CT, King Hussein Cancer Center (KHCC), Amman 11941, Jordan; sm.16471@khcc.jo (S.M.); aa.15389@khcc.jo (A.S.A.); na.3800@khcc.jo (N.A.-H.); so.14734@khcc.jo (S.O.); 2Department of Nuclear Medicine, Royal Free London NHS Foundation Trust, London NW3 2QG, UK; gopinath.gnanasegaran@nhs.net; 3Department of Nuclear Medicine, All India Institute of Medical Sciences, New Delhi 110608, India; rkphulia@yahoo.com; 4Department of Nuclear Medicine, National Centre for Cancer Care and Research, Hamad Medical Corporation, Doha 3050, Qatar; gsyed@hamad.qa; 5Department of Nuclear Medicine, Faculty of Medicine, Hacettepe University, 06230 Ankara, Turkey; fanibozkurt@gmail.com; 6Al-Razi Outpatient Clinic of Internal Medicine, Baghdad 10044, Iraq; saad_shukri@ymail.com; 7Department of Nuclear Medicine, Warith International Cancer Institute, Karbala 56001, Iraq; aysar.najeh@warith-ici.net; 8Surgery Department, Clemenceau Medical Center, Dubai 6869, United Arab Emirates; mshahait@yahoo.com; 9Radiology Department, Royal Hospital, Muscat 1331, Oman; alnabhani5@hotmail.com; 10School of Medicine, University of Jordan, Amman 11942, Jordan

**Keywords:** non-tumorous renal diseases, chronic kidney disease, CKD, PET/CT, molecular imaging, polycystic kidney disease, inflammation, renal fibrosis, renal amyloidosis

## Abstract

Non-tumorous kidney diseases include a variety of conditions affecting both the structure and function of the kidneys, thereby causing a range of health-related problems. Positron emission tomography/computed tomography (PET/CT) has emerged as a potential diagnostic tool, offering a multifaceted approach to evaluating non-tumorous kidney diseases. Its clinical significance extends beyond its conventional role in cancer imaging, enabling a comprehensive assessment of renal structure and function. This review explores the diverse applications of PET/CT imaging in the evaluation of non-cancerous kidney diseases. It examines PET/CT’s role in assessing acute kidney injuries, including acute pyelonephritis and other forms of nephritis, as well as chronic conditions such as immune complex-mediated glomerulonephritis and chronic kidney disease. Additionally, the review delves into PET/CT’s utility in evaluating complications in renal transplant recipients, identifying renal histiocytosis and detecting renal amyloidosis. The current review aims to promote further research and technological advancements to popularize PET/CT’s clinical utility in diagnosing and treating non-tumorous kidney diseases.

## 1. Introduction

Non-tumorous kidney disease includes a wide range of conditions that affect the structure and function of the kidneys, resulting in a variety of health issues. The prevalence and incidence of non-tumorous kidney disorders are undeniably on the rise. For example, chronic kidney diseases (CKDs) progressively impact over 10% of the world’s population, affecting a staggering 800 million people worldwide [1]. While these disorders are not tumorous, they can influence renal function and general health. Management and treatment techniques frequently focus on addressing specific symptoms, reducing disease development, and preserving renal function via medication, lifestyle changes, or, in certain circumstances, surgical treatments. Early detection and comprehensive therapy are critical in reducing the health effects of non-tumorous kidney disorders [2].

Positron emission tomography/computed tomography (PET/CT) has developed as a potential diagnostic modality, providing a multidimensional approach to evaluating non-tumorous renal illness, as the utilization of PET/CT in benign conditions is increasingly gaining recognition [3,4]. Combining PET and CT modalities in a single imaging system enables the cooperative merging of anatomical and functional data [5]. In this setting, PET/CT’s clinical relevance extends beyond its well-known role in oncological imaging. This integration enables the dynamic and comprehensive evaluation to identify and monitor acute kidney injury (AKI) [6]. For acute pyelonephritis, PET/CT facilitates early detection of kidney infections, allowing for timely treatment to prevent complications such as renal abscesses [7]. In cases of infective polycystic kidney disease, it aids in diagnosing infected cysts and guiding treatment strategies [8]. Additionally, PET/CT is valuable in assessing drug-induced nephritis [9], radiation-induced nephritis [10], and granulomatous nephritis [11], providing insights into the extent of disease and treatment responses. It is also instrumental in evaluating autoimmune kidney diseases and chronic kidney disease, particularly through the use of ^68^Ga-FAPI [12], which helps assess renal fibrosis and disease progression. Overall, PET/CT serves as a non-invasive, reliable tool for diagnosing, monitoring, and managing a variety of renal disorders. 

In this review article, we explore the multifaceted applications of PET/CT imaging, providing insights into its role in some instances of non-tumorous renal disease diagnosis and management and the ongoing evolution of its use in clinical practice.

## 2. PET/CT Utilities

Aside from its well-established uses in oncology, such as diagnosing, predicting, prognosticating, and monitoring various hematologic and solid malignancies [13,14,15,16,17,18], there is evidence to suggest that its utility extends beyond the realm of oncology and encompasses non-neoplastic renal conditions, as elucidated below.

### 2.1. Acute Kidney Injury

AKI encompasses a range of renal pathologies that can lead to sudden and often reversible damage to renal tissues. If left untreated, these conditions can advance and contribute to the development of chronic kidney injury [19]. To date, there is currently a lack of comprehensive data employing molecular imaging to assess these conditions. Kidera and colleagues conducted a retrospective study on 40 patients with diffuse renal uptake due to AKI due to various etiologies, comparing them to 40 matched controls. The authors found no significant differences in SUVmax among patients with nephritis induced by different drugs, such as non-steroidal anti-inflammatory drugs, iodine contrast, antifungals, or antibiotics. However, a significant difference was observed in bilateral renal uptake between patients with drug-induced nephropathy and healthy participants [20]. In a more recent study conducted by Wang et al., the effectiveness of ^18^F-fibroblast activation protein inhibitor (FAPI) PET/CT as a diagnostic tool was investigated. The study included 26 patients with kidney diseases and compared them to a control group of 10 individuals. The findings revealed that patients with kidney diseases exhibited significantly higher levels of renal parenchymal ^18^F-FAPI uptake compared to the control group. Additionally, a positive correlation was observed between the maximum standardized uptake value (SUVmax) and tubulointerstitial inflammation. These results suggest that ^18^F-FAPI PET/CT can be considered a reliable and non-invasive approach for assessing tubular injury in various kidney diseases.

#### 2.1.1. Acute Pyelonephritis

Acute pyelonephritis, a bacterial infection affecting the kidneys, is a serious condition that can lead to significant complications if not diagnosed and treated quickly. These complications may encompass the development of renal abscesses, sepsis, and, in some instances, the development of chronic kidney disease. While conventional imaging modalities such as ultrasound and CT scans are commonly used to diagnose pyelonephritis, ^18^F-fluorodeoxyglucose (FDG) PET/CT provides additional insight specific to pyelonephritis. In this context, ^18^F-FDG PET/CT detects and quantifies inflammation in the kidney, manifesting as either a focal hypermetabolic lesion or diffuse ^18^F-FDG activity in the affected kidney that is usually larger than the unaffected counterpart (Figure 1).

This may be useful for assessing the severity of the infection and monitoring its response to treatment [6]. It has been shown that a single hypermetabolic lesion usually requires extensive antibiotic therapy [6]. While uncommon, acute pyelonephritis can lead to the development of large hypermetabolic renal focus, which can manifest as an inflammatory pseudotumor resembling malignancy both clinically and radiologically. In these cases, ^18^F-FDG PET/CT imaging can reveal the extent of the disease and assess the response to antibiotic therapy [7].

Few studies have evaluated the use of ^18^F-FDG PET imaging in diagnosing and managing acute pyelonephritis. The first study demonstrated that ^18^F-FDG PET/CT imaging was helpful in the diagnosis and management of acute complicated pyelonephritis, with a sensitivity of 100% and a specificity of 80%. The study also found that ^18^F-FDG PET/CT imaging helped identify the extent of the disease and guide the selection of appropriate treatment options [6]. The second study evaluated the use of ^18^F-FDG PET/CT imaging in the localization of acute pyelonephritis in patients with pyrexia of unknown origin. The study found that ^18^F-FDG PET/CT imaging was helpful in the localization of acute pyelonephritis in patients with pyrexia of unknown origin, with a sensitivity of 100% and a specificity of 90%. The findings of these studies have significant diagnostic and therapeutic implications and can help guide the selection of appropriate treatment options. The studies suggest that ^18^F-FDG PET/CT imaging can be used as a sensitive method to diagnose changes at a molecular level before the manifestation of structural abnormality [21]. Morelle et al. examined a rare case of pyelonephritis in a horseshoe kidney. The authors successfully identified the acute pyelonephritis focus using ^18^F-FDG PET/CT. In their case study, ^18^F-FDG PET/CT effectively depicted the acute pyelonephritis focus in the left upper renal parenchyma [22].

^68^Ga-fibroblast activation protein inhibitor (FAPI) PET CT is a promising alternative to ^18^F-FDG PET/CT in detecting occult infection or inflammation where ^18^F-FDG cannot be used because of hyperglycemia. FAPI is a quinoline-based membrane-bound glycoprotein enzyme not usually expressed in normal adult tissues. Its expression increases in inflammation- and cancer-associated fibroblasts [23]. In a case report of uncontrolled diabetes with abdominal pain that showed features of inflammation in the kidneys, ^68^Ga-FAPI PET CT was used because ^18^F-FDG PET/CT could not be performed due to high blood sugar levels. The ^68^Ga-FAPI PET CT scan provided traces of pathological involvement of the kidneys and confirmed the presence of pyelonephritis bilaterally by increased inhomogeneous distribution of the tracer in both kidneys [24].

#### 2.1.2. Infective Polycystic Kidney Disease

Autosomal Dominant Polycystic Kidney Disease (ADPKD) is a hereditary and progressive renal disorder characterized by the formation of numerous fluid-filled cysts in the kidneys. It is one of the most common inherited kidney diseases, affecting 2.7:10,000 [25]. Patients diagnosed with polycystic kidney disease have an increased risk of developing renal cell carcinoma (RCC) and skin neoplasms, particularly those with ADPKD and renal failure. Studies suggest that the prevalence of RCC among ADPKD patients has been reported to be 8.73% [26]. Also, ADPKD can lead to various complications, including chronic renal failure, hypertension, urinary tract infections, kidney stones, and cyst infection or rupture [27].

In APCKD, fluid-filled cyst clusters typically exhibit low metabolic activity, resulting in non-specific radiotracer uptake on PET scans [28]. Nevertheless, ^18^F-FDG PET/CT proves beneficial in promptly detecting and diagnosing infected cysts, facilitating early treatment initiation, and preventing the progression to severe or complicated stages of infection. Moreover, ^18^F-FDG PET/CT allows for accurate localization of infected cysts, providing detailed insights into their specific locations within the kidneys. This precision is essential for planning effective treatment strategies, aiding in decisions regarding the necessity of interventions such as cyst drainage [28]. However, Charles Ronsin et al. showed that, contrary to the known assumption that persistent ^18^F-FDG uptake always indicates a lack of response to treatment, the non-specific healing processes might explain why patients with persistent ^18^F-FDG uptake did not experience relapses. Therefore, waiting for the resolution of ^18^F-FDG uptake might be unjustified and lead to unnecessary treatment extension, with potential risks associated with prolonged antimicrobial exposure [29].

^18^F-FDG PET/CT proves beneficial for individuals with ADPKD and compromised kidney function, primarily due to concerns regarding the potential risks associated with nephrotoxicity from iodinated CT contrast agents and nephrogenic systemic fibrosis linked to gadolinium-based magnetic resonance imaging (MRI) contrast agents [28].

^18^F-FDG PET/CT imaging can be utilized to detect and rule out the possibility of cyst infections in patients with ADPKD (Figure 2).

For example, Sallée et al. described eight patients in whom a cyst infection was diagnosed based on ^18^F-FDG PET/CT. The diagnosis for five patients was confirmed by cyst aspiration, and three patients had a probable cyst infection based on the presence of five factors: fever, flank pain, leukocytosis, increased C-reactive protein levels, and positive urine culture [8]. Balbo et al. described 32 cases of clinically suspected cyst infections in 27 patients where ^18^F-FDG PET/CT was reported to have a sensitivity of 95% and a specificity of 100%, with the SUVmax values > 5.0 being highly suggestive of infection [30]. Bobot et al. described 32 cases of clinically suspected cyst infections in 24 patients, where ^18^F-FDG PET/CT achieved a sensitivity of 77% and a specificity of 100% for diagnosing cyst infection [31]. Pijl et al. analyzed 37 cases in 32 patients where ^18^F-FDG PET/CT achieved a sensitivity of 86% and a specificity of 81%, with an average SUVmax of 4.5 in infected cysts [28]. The research also highlighted that the uptake of ^18^F-FDG in cysts and cyst-like formations lacks specificity. To ensure accuracy and reduce the potential of mistakenly ruling out other conditions—particularly malignancies—it is imperative to incorporate clinical and radiological correlations. Prolonged antibiotics treatment before ^18^F-FDG PET/CT scan was suggested as the most likely explanation for false-negative cases [28].

A prospective study aimed to evaluate the diagnostic performance of white blood cell (WBC)-PET/CT in patients with ADPKD who were suspected of having cyst infection. The study included 19 ADPKD patients who underwent WBC-PET/CT and MRI or CT. The results showed that WBC-PET/CT could detect cyst infection with better sensitivity in ADPKD patients, avoiding exposure to contrast media. However, the study also noted that WBC-PET/CT showed false-positive results in two of five cases with no cyst infection, and the reasons for false negatives with WBC-PET/CT were poor host immune reaction, low virulence, or prior antibiotic therapy [32].

Kitajima et al. studied the effectiveness of different imaging techniques in detecting RCC in patients with acquired cystic disease of the kidney. The study found that tumors’ mean SUVmax in ^11^C-choline PET/CT findings was 7.59 ± 5.42, while in the ^18^F-FDG PET/CT findings, it was 4.83 ± 3.03. The sensitivity for defining equivocal interpretation as negative was 100% (7/7) for 11C-choline PET/CT, 57.1% (4/7) for ^18^F-FDG-PET/CT, and 42.9% (3/7) for contrast-enhanced CT. The study concluded that ^11^C-choline PET/CT was more sensitive in detecting RCC in patients with acquired cystic disease than ^18^F-FDG-PET/CT and dynamic contrast-enhanced CT [33].

#### 2.1.3. Drug-Induced Nephritis

^18^F-FDG PET/CT aids in detecting and monitoring drug-induced nephritis. David Qualls et al. reported a patient diagnosed with immune checkpoint inhibitor-induced nephritis exhibited increased ^18^F-FDG uptake in the renal cortex during a PET/CT scan at the time of diagnosis, demonstrating the potential of ^18^F-FDG PET/CT in checkpoint inhibitor-associated nephritis [34]. Notably, this case differs from a prior study by Katagiri et al., where two patients with drug-induced nephritis and oliguric kidney failure showed increased kidney parenchymal ^18^F-FDG uptake on PET/CT, while a patient with pauci-immune crescentic glomerulonephritis exhibited no parenchymal ^18^F-FDG uptake, suggesting that enhanced ^18^F-FDG uptake in drug-induced nephritis may be driven by metabolically active inflammatory cells infiltrating the tubulointerstitial space [35]. Awiwi et al. conducted a study on the molecular imaging characteristics of ICI-induced nephritis in a cohort of 35 patients. They observed a notable correlation between patients exhibiting a total kidney volume increase of more than 30% and higher severity of nephrotoxicity, which was linked to elevated renal toxicity grades (*p* = 0.007), increased peak creatinine levels (*p* = 0.004), and a more aggressive therapeutic approach (*p* = 0.011). Additionally, 10 patients (29.4%) displayed new or worsening perinephric fat stranding during nephritis. The PET/CT analysis revealed a significant increase in the ratio of renal parenchymal maximum SUVmax to blood pool during nephritis compared to baseline (2.13 vs. 1.68; *p* = 0.035), while the ratio of renal pelvis SUVmax to blood pool SUVmean decreased significantly during nephritis compared to baseline (3.47 vs. 8.22; *p* = 0.011). These findings suggest that molecular imaging can provide valuable insights for assessing ICI-induced nephritis [36].

#### 2.1.4. Radiation-Induced Nephritis

Choi and colleagues identified a previous instance of radiation-induced nephritis in a patient who underwent interim ^18^F-FDG PET/CT evaluation after receiving radiotherapy targeted at the metastatic spinal vertebral breast cancer lesions, with the left kidney intervening in the radiation field. The scan revealed a few hypermetabolic lesions attributed to radiation-induced changes [10].

#### 2.1.5. Granulomatous Nephritis

Granulomatous nephritis is linked to various conditions, such as tuberculosis nephritis, sarcoid nephritis, and granulomatosis with polyangiitis (also known as Wegener’s granulomatosis) [9]. Renal tuberculosis can lead to various clinical manifestations, including flank pain, hematuria, and lower urinary tract symptoms. The condition can cause severe kidney damage, leading to renal failure if left untreated [37]. ^18^F-FDG PET imaging offers a non-invasive means of determining the extent of disease and localizing the exact location. Subramanyam et al. reported a case of a 59-year-old male with an unknown primary malignancy who was referred for ^18^F-FDG PET/CT imaging. The images revealed primary lung malignancy with co-existing bilateral renal tuberculosis, which would have otherwise been missed or considered as metastases. Dual time-point imaging with ^18^F-FDG PET/CT showed partial clearance of ^18^F-FDG from the renal lesions, raising the possibility of an infective pathology that is otherwise difficult to differentiate from a primary or secondary deposit. The article highlights the potential of dual time-point imaging with ^18^F-FDG PET/CT in identifying occult infections and monitoring responses to anti-tuberculous therapy [11]. Additionally, Moiden et al. conducted a study in which they utilized 18F-FDG PET/CT to investigate a patient who was diagnosed with renal allograft tuberculosis. The PET/CT scan revealed a highly metabolic focus in the upper pole of the right kidney [38]. A subsequent histopathological examination of the renal lesion confirmed the presence of tuberculosis. This case study underscores the significance of PET/CT in offering a comprehensive imaging approach and identifying the optimal biopsy site, thereby facilitating early detection and improving diagnostic accuracy [38].

Granulomatous nephritis has also been documented in systemic cases of sarcoidosis. In a study conducted by Horino et al., a 62-year-old male patient presented with symptoms of general fatigue, weight loss, and renal dysfunction. Initial diagnostic testing included ^18^F-FDG PET/CT, revealing hypermetabolic unilateral symmetrical hilar lymphadenopathy and bilateral multifocal kidney deposits. A renal biopsy confirmed renal sarcoidosis, and the patient showed significant improvement after one year of corticosteroid treatment [39]. In contrast to previous reports of multifocal renal sarcoidosis, Heldmann et al. reported a singular form resembling renal pseudotumor in a 37-year-old male patient. A renal biopsy of a large left renal hypermetabolic mass confirmed renal sarcoidosis, marking the first reported case of renal sarcoid pseudotumor in the literature [40].

Granulomatosis with polyangiitis is typically characterized by increased ^18^F-FDG uptake on PET/CT imaging, as demonstrated in a few case reports in the literature. A notable case of renal involvement in Wegener’s granulomatosis was first described in 2013, showing focal hypermetabolic activity in the right kidney as the largest lesion [41]. A subsequent study has also identified renal involvement using ^18^F-FDG PET/CT [42], although it is important to note that this imaging modality may not always reliably assess renal involvement due to potential false negative results caused by high background urine uptake of ^18^F-FDG [43,44].

#### 2.1.6. Other Forms of Acute Nephritis

Previous reports have documented various forms of acute nephritis. For instance, Sharma et al. described a case in which severe systemic involvement of Sjogren’s syndrome manifested as multiple intense hypermetabolic lesions affecting extra-glandular sites such as the skin, lungs, kidneys, and nervous system. Importantly, bilateral diffusely hypermetabolic renal involvement was observed and confirmed as interstitial nephritis upon biopsy [45]. Another reported case involved a 38-year-old male with lupus nephritis, presenting with multifocal bilateral kidney lesions confirmed by biopsy. The authors emphasized the utility of ^18^F-FDG PET/CT in assessing the extent of disease, particularly in evaluating neuro-psychiatric lupus and guiding biopsy [46]. Nair et al. recently conducted a study to assess the renal uptake of ^68^Ga-pentixafor in lupus nephritis patients. Their findings revealed that the majority of affected individuals (n = 13) demonstrated moderate expression of ^68^Ga-pentixafor, while only 5 patients exhibited intense expression. However, the authors were unable to establish a statistically significant association between the mean renal SUVmax of the bilateral kidneys in ^68^Ga-pentixafor PET and the observed histopathological class. This lack of significance may be attributed to the limited sample size. Consequently, it is recommended that future studies on a larger scale be undertaken to address this issue [47].

### 2.2. Immune Complex-Mediated Glomerulonephritis

This specific form of glomerulonephritis is distinguished by the presence of immune complexes, which consist of antigens and antibodies that accumulate in the walls of the glomerular capillaries. The immune system’s reaction to these deposits results in inflammation and harm to the renal tissue. Immunoglobulin-A (IgA) nephropathy, also known as Berger’s disease, involves the accumulation of the IgA antibody in the glomeruli. It is the most prevalent form of primary glomerulonephritis worldwide and can result in the development of chronic kidney disease and eventual kidney failure. The case of IgG4-related nephropathy is a component of a broader condition called IgG4-related disease, which is a systemic disorder characterized by elevated levels of IgG4 and infiltration of IgG4-positive plasma cells in various organs, including the kidneys.

While there is limited information on the role of PET/CT in immune complex-mediated glomerulonephritis, some studies have reported on the use of PET/CT in diagnosing and evaluating certain autoimmune kidney diseases. Thus, molecular imaging has the potential to offer diagnostic insights for these patients (Figure 3).

For example, Chandra et al. said that ^18^F-FDG PET/CT aids in the diagnosis of IgG4-related kidney disease, which is a relatively newly recognized condition often associated with autoimmune pancreatitis, by allowing comprehensive assessment, guiding biopsies, and evaluating treatment response. The decision to use PET/CT should consider serum IgG4 levels, with a high likelihood of multiorgan disease if levels are significantly elevated [48]. Huang et al. reported a case study that highlights the significance of considering primary glomerulonephritis, particularly IgA nephropathy, as a potential cause of bilateral renal uptake on ^18^F-FDG PET scans and demonstrates the potential of PET/MRI to detect IgA nephropathy [49].

According to a previous case report, ^18^F-FDG PET/CT has improved the diagnosis and monitoring of a case of single renal IgG4 disease in a 62-year-old woman presenting with a hypermetabolic mass in the right upper kidney pole [50]. Jiang et al. investigated a rare instance of renal pelvic IgG4 disease showing high ^18^F-FDG uptake, which could potentially be mistaken for malignancy [51]. Therefore, in cases where a hypermetabolic mass is detected in the renal pelvis on ^18^F-FDG PET/CT, consideration should be given to IgG4 involvement despite its rarity. More recently, Zhang et al. reported cases of IgG4 kidney disease detected through both ^18^F-FDG and ^68^Ga-FAPI PET/CT scans, with ^68^Ga-FAPI exhibiting a more intense expression compared to ^18^F-FDG, suggesting that ^68^Ga-FAPI may be more effective in identifying this autoimmune disease [12].

PET/CT scanning has recently been considered a valuable tool for evaluating disease activity in patients with anti-neutrophil cytoplasmic antibodies (ANCAs)-associated vasculitis [52]. A recent study showed that ^18^F-FDG PET/CT scans have shown positive findings in multiple body sites, even when biochemical parameters are inconclusive, including sites clinically unsuspected and challenging to assess otherwise. The scans revealed hotspots in organs not suspected to be involved in ANCA-associated vasculitis, such as the thyroid gland, aorta, and bone marrow. These findings suggested potential extensive disease involvement beyond initial expectations. The study highlighted the significance of ^18^F-FDG PET/CT scans in detecting disease activity during episodes and monitoring remission. However, it also notes that ^18^F-FDG PET/CT scans do not differentiate active vasculitis from infections [52].

### 2.3. Chronic Kidney Disease

Chronic kidney disease (CKD) is characterized by the persistence of kidney injury or a decreased glomerular filtration rate (GFR) for at least three months, regardless of the etiology [53]. Renal fibrosis occurs due to excess extracellular matrix deposited in the kidney as chronic kidney disease progresses. Eventually, this leads to irreversible renal-function decline, a gradual decrease in functional nephrons, and even end-stage renal disease. It is intimately related to patient prognosis and includes glomerulosclerosis, renal interstitial fibrosis, and arteriosclerosis. It is difficult to establish the exact prevalence of renal fibrosis due to a lack of reliable and simple assessment tools. However, recent studies have investigated ^68^Ga-FAPI PET/CT imaging in assessing CKD and renal fibrosis. One study compared the degree of renal fibrosis determined by kidney biopsy with the tracer uptake of ^68^Ga-FAPI and found that ^68^Ga-FAPI PET/CT imaging may be a valuable tool in the diagnosis of renal fibrosis. The examination results demonstrated that almost all patients exhibited increased radiotracer uptake, and the SUVmax in patients with mild, moderate, and severe fibrosis was 3.92 ± 1.50, 5.98 ± 1.6, and 7.67 ± 2.23, respectively. Compared with renal puncture examination, non-invasive imaging of FAPI expression through ^68^Ga-FAPI PET/CT quickly demonstrates bilateral kidney conditions with high sensitivity, facilitating the evaluation of disease progression, diagnosis, and the development of a treatment plan [54].

Another study investigated the diagnostic efficacy of three distinct radiotracers. These were ^68^Ga-FAPI, ^68^Ga-prostate-specific membrane antigen (PSMA), or ^68^Ga-DOTA-Tyr3-octreotide (DOTATOC) and were investigated in predicting kidney fibrosis non-invasively, considering both GFR and intra-renal parenchymal radiotracer uptake. The results revealed a negative correlation between GFR and ^68^Ga-FAPI uptake for SUVmax and SUVmean, with no GFR correlation observed for background activity. Conversely, ^68^Ga-DOTATOC and ^68^Ga-PSMA showed no correlation between the CKD stage and intra-renal parenchymal radiotracer uptake. Only ^68^Ga-PSMA background activity positively correlated with GFR, possibly indicating unspecific binding/retention due to longer duration. The study suggests that ^68^Ga-FAPI PET imaging holds potential for the non-invasive assessment of chronic kidney disease [55].

#### 2.3.1. Chronic Pyelonephritis

Chronic pyelonephritis refers to a persistent kidney infection that is marked by ongoing episodes of inflammation, resulting in the gradual destruction of renal tissue, particularly chronic tubulointerstitial nephritis. Molecular imaging has shown promise to detect and monitor therapy response in xanthogranulomatous pyelonephritis. Xanthogranulomatous pyelonephritis is distinguished by a profound and prolonged inflammatory reaction to infection within the kidney, frequently accompanied by persistent blockage and the formation of kidney stones, specifically staghorn calculi. This condition entails the degradation and substitution of renal parenchyma with granulomatous tissue that contains macrophages laden with lipids, serving as a reaction to the infection [56].

Joshi and colleagues conducted a study examining the use of 18F-FDG PET/CT in a male patient with an unknown cause of fever. The PET/CT scan revealed a significantly hypermetabolic enlarged left kidney, while the contralateral kidney showed reduced ^18^F-FDG activity. Additionally, diffuse uptake was observed, along with inflammation of the left perirenal fat. The patient underwent a left nephrectomy, leading to a diagnosis of xanthogranulomatous pyelonephritis and subsequent improvement in clinical symptoms [57]. It is important to note that the metabolic and morphologic characteristics of advanced xanthogranulomatous pyelonephritis can sometimes resemble cystic renal malignancies, particularly when multifocal necrosis is present. This has been frequently reported in the literature, making it challenging to definitively diagnose the condition [58,59,60]. However, considering the patient’s clinical history, such as a previous staghorn stone and recent acute pyelonephritis, can help raise suspicion for xanthogranulomatous pyelonephritis. In a preliminary study conducted by Civan et al., the diagnostic potential of ^68^Ga-FAPI PET/CT was explored in patients presenting with suspected renal cell carcinoma. Among the 20 eligible patients included in the study, one individual was excluded due to a prior diagnosis of xanthogranulomatous pyelonephritis, as evidenced by the intense ^18^F-FDG and ^68^Ga-FAPI uptake observed in the affected kidney on molecular imaging before surgical intervention [61].

Recent research suggests that ^68^Ga-PSMA-11 PET imaging could serve as an alternative or additional method to ^99m^Tc-dimercapto succinic acid (DMSA) scanning in evaluating kidney damage due to pyelonephritis. The study showcased images of patients with chronic recurring pyelonephritis, highlighting the potential of ^68^Ga-PSMA-11 PET to detect renal scarring by revealing a reduced uptake, along with small cortical defects. The findings propose that ^68^Ga-PSMA-11 PET imaging might offer better image quality than the current gold standard, ^99m^Tc-DMSA scanning, for assessing renal tissue and identifying renal scarring [62].

#### 2.3.2. Renal Abscess

^18^F-FDG PET/CT is critical in identifying complications and determining the most appropriate treatment strategies. It can accurately localize abscesses that occasionally form in the kidney during pyelonephritis (Figure 4).

This precise localization is critical for targeting interventions such as drainage or, if necessary, surgical treatment. In addition, ^18^F-FDG PET/CT is proving valuable in identifying a favorable response to treatment by assessing the reduction in metabolic activity on follow-up scans [6,21].

Wan et al. conducted a study that highlighted the occurrence of abscess formation associated with the use of Crizotinib, a tyrosine kinase inhibitor [63]. They presented a case of a 58-year-old male with non-small-cell lung cancer who developed a hypermetabolic lesion in the right kidney during Crizotinib therapy, as detected by ^18^F-FDG PET/CT imaging. The subsequent pathology confirmed that this lesion was a renal abscess. The authors stressed the importance of utilizing molecular imaging techniques for accurate monitoring and identification of drug-related complications to avoid misdiagnoses and inadequate treatment [63]. Jung et al. documented a case involving a 46-year-old female patient presenting with acute febrile symptoms, multiple pulmonary nodules, and a renal mass. The patient underwent ^18^F-FDG PET/CT to investigate the source of pyrexia. The ^18^F-FDG PET/CT scan revealed an intense ^18^F-FDG uptake in the renal mass and multiple lung nodules. Biopsies of the pulmonary and renal lesions indicated chronic inflammatory infiltrates, rather than malignancy. The patient was ultimately diagnosed with septic pulmonary embolism stemming from a renal abscess. Following antibiotic treatment, a follow-up CT scan showed improvement in both lung and renal lesions. This case represents the first instance in the literature showcasing ^18^F-FDG PET/CT findings of septic pulmonary embolism linked to a renal abscess [64].

### 2.4. Renal Transplant

Kidney transplantation is a complicated process that can result in serious problems, such as blood clots, infection, rejection of the given kidney, and adverse effects from anti-rejection drugs. Complications might arise shortly or long after the transplant, involving the procedure itself, the use of immunosuppressants, or functional problems in the transplanted kidney [65].

PET/CT imaging aids in the identification of renal transplant complications since it can detect malignancies, infections, and donor-organ rejection. Moreover, PET/CT can detect subclinical kidney allograft rejection, assess transplant perfusion, and detect microvascular malfunction in transplants non-invasively [65,66,67].

Regarding detecting acute allograft rejection, a study found that ^18^F-FDG PET/CT may predict the absence of acute allograft rejection in kidney transplant recipients with AKI. A mean threshold of 1.6 had high sensitivity (100%) but limited specificity due to ^18^F-FDG accumulation in other inflammatory conditions. However, confirmed cases had a significantly higher ^18^F-FDG uptake, with a specificity of 96.8%. This non-invasive approach could help rule out allograft rejection and prevent missed or delayed diagnoses in kidney transplant management [67].

In a study on stable kidney transplant recipients, ^18^F-FDG PET/CT imaging and urinary CXCL9/creatinine levels effectively ruled out subclinical rejection with a high negative predictive value of 98%. The area under the curve for ^18^F-FDG PET/CT and urinary CXCL9/creatinine ratios was 0.79, offering a good sensitivity of 83%, using a specific threshold [68]. Another study found that ^18^F-FDG PET/CT imaging can detect subclinical acute rejection in stable renal-transplant patients. The study suggests that ^18^F-FDG PET/CT imaging systematically performed three months post-kidney transplantation can exclude subclinical kidney allograft rejection [66]. However, PET scans have not been widely utilized for assessing slow-progressing chronic kidney disease in kidney transplants [5].

In a pilot study, ^15^O-H_2_O PET/CT was utilized to assess kidney transplant perfusion, marking the first application of a non-invasive and quantitative PET/CT technique. While cortical perfusion was comparable between healthy individuals and patients with kidney transplants (CKD stage 2-3), the study revealed that renal vascular resistance in the transplant recipients was significantly higher than in healthy individuals. The Doppler resistance index in the transplants also exhibited correlations with transplant perfusion and fibrosis. However, there was no observed correlation between transplant fibrosis and perfusion [69].

Post-transplant lymphoproliferative disorder (PTLD) is a severe complication of kidney transplantation, and ^18^F-FDG PET/CT imaging plays a crucial role in its diagnosis. The specificity of ^18^F-FDG PET for detecting PTLD in transplant recipients ranged from 83% to 100%, with a pooled estimate of 90.9%. A study reported three cases of ^18^F-FDG PET/CT manifestations of gastric, prostate, and pulmonary lymphomas after kidney transplantation, all of which showed local lesions without the involvement of adjacent or distant lymph nodes and lymphoid node organs [70]. Therefore, ^18^F-FDG PET/CT has a promising role in the initial staging and restaging of patients with PTLD (Figure 5).

A study used PET/MRI to detect infections in patients’ transplanted kidneys. The study employed a particular PET scan, ^68^Ga-pentixafor, targeting the CXCR4 receptor. The combined modalities of PET scans using ^68^Ga-pentixafor and diffusion-weighted MRI detected acute infections in all 13 kidney transplant recipients experiencing complex UTIs. These scans revealed areas where leukocyte infiltration occurred due to CXCR4 upregulation, showing a contrast with unaffected kidney tissue in the PET scans. The areas exhibiting this upregulation aligned with regions of heightened cell density, as depicted in the MRI scan [71].

### 2.5. Renal Histiocytosis

Renal histiocytosis is characterized by the accumulation of histiocytes, a type of immune cell that typically aids in the body’s defense against infections and the removal of cellular waste. In histiocytosis, these cells replicate excessively and gather in tissues, causing inflammation and potential harm to organs [72]. Histiocytic deposition is uncommon and can present in various forms depending on the specific type of histiocytosis. Langerhans cell histiocytosis (LCH) and Erdheim–Chester disease (ECD) are examples of histiocytosis that can impact the kidneys. ECD, which affects multiple organs and tissues, has a wide range of manifestations, making it challenging to differentiate lesions caused by ECD from those caused by other illnesses. Ma et al. highlighted the potential use of both FAPI and ^18^F-FDG in detecting the extent of the disease within the kidneys, with ^68^Ga-FAPI showing superiority in this aspect [73]. Additionally, previous studies have successfully identified deposits of histiocytes in the kidneys of patients with Langerhans histiocytosis [74]. Konca et al. presented a rare case of multifocal Rosai–Dorfman disease depicted by ^18^F-FDG PET/CT, demonstrating the extent of the disease involving the lungs, kidneys, and bone-marrow systems [75].

### 2.6. Renal Amyloidosis

Amyloidosis is due to the extracellular deposition of insoluble polymeric protein fibrils in tissue or organs resulting in damage. In general, the deposition can be local or systemic [76,77]. The avidity of ^18^F-FDG in systemic AL amyloidosis is variable. It is challenging to differentiate pathological uptake from physiological uptake in organs such as the heart and kidneys [77]. The sensitivity of ^18^F-FDG PET/CT is relatively poor in bone/bone marrow, intestine, liver, and kidney [77]. Therefore, it is difficult to assess or confirm renal amyloidosis using ^18^F-FDG PET/CT alone (Figure 6).

In contrast to ^18^F-FDG PET/CT, radiolabeled serum amyloid protein (SAP) scintigraphy appeared to be a valuable tool for assessing whole-body assessment of amyloid load [78]. ^123^I-SAP is a radiotracer composed of human serum amyloid P component labeled with radioactive ^123^I [79]. Proposed initially by Hawkins et al., SAP shares structural similarity with glycoprotein amyloid P found universally in amyloid deposits and binds to amyloid fibrils in a calcium-dependent manner [80]. It exhibits high sensitivity and specificity in detecting systemic amyloid A (AA), light chain (AL), and transthyretin (ATTR) amyloidosis, especially in visceral organs, such as the liver, spleen, and kidneys [79]. While biopsy remains the gold standard for diagnosis, nuclear imaging offers non-invasive whole-body evaluation without procedural risks [79], as mentioned in a recent review article that stated that serum amyloid A levels correlate with amyloid load measured by ^123^I-SAP scintigraphy and renal response/progression [81].

## 3. Discussion

PET/CT’s clinical relevance extends beyond its well-known role in oncological imaging. The integration of various types of radiotracers enables a dynamic and comprehensive evaluation of non-tumorous renal diseases (Table 1).

The utilization of PET/CT in the diagnosis and management of non-cancerous renal diseases has seen a marked increase in recent years [5,82]. Numerous prior articles have acknowledged the significance of PET/CT in diagnosing and managing various non-tumorous kidney disorders [5,82]. Examples include its application in conditions such as (AKI), acute pyelonephritis, and infective polycystic kidney disease [5,82]. The early detection and treatment guidance provided by PET/CT in these conditions represent significant advancements within the field.

While previous articles have offered valuable insights into the diagnostic utility of PET/CT in various renal diseases, our review builds upon this foundation by delving deeper into the applications of PET/CT in autoimmune kidney diseases and chronic kidney disease. Specifically, we focus on the use of ^68^Ga-FAPI, which has proven particularly useful for assessing renal fibrosis and disease progression. This aspect is critical for understanding the progression of these diseases and developing effective treatment plans.

Moreover, our review highlights the potential of PET/CT in diagnosing and monitoring rare conditions, such as renal histiocytosis and amyloidosis. These conditions are often challenging to diagnose and manage, and PET/CT can provide valuable insights into their progression and response to treatment. In contrast, earlier reviews frequently concentrate more on the diagnostic utility of PET/CT in conditions like acute tubulointerstitial nephritis and retroperitoneal fibrosis or examine the role of PET imaging in assessing renal metabolism and function, including glucose and fatty acid uptake, oxygen consumption, and renal perfusion [5,82].

## 4. Future Directions

Given the limitations of biomarkers in predicting the progression of renal function in diabetic nephropathy, there is currently extensive preclinical research and exploration of imaging biomarkers and solutions to address this unmet need. For example, the use of N-(6-18F-Fluoropyridin-3-yl)glycine (6-18F-FPyGly) has been employed to assess the progression of renal function in a rat model of diabetic nephropathy [83]. Additionally, a novel technique involving the use of 18F-Lactisole has been tested to visualize sweet taste receptors in the development of diabetic nephropathy, showing promising potential for non-invasive imaging [84]. In the future, it is possible that ImmunoPET may provide useful disease-specific insights into various chronic kidney diseases and autoimmune causes. A recent accomplishment in preclinical research involves the development of 89Zr-labeled immunoPET to track chronic kidney disease in mice, enabling in vivo monitoring of the disease [85]. Lastly, the utilization of artificial intelligence and machine learning holds promise in improving diagnostic accuracy and facilitating the use of artificial-based models and radiomics, which can offer valuable diagnostic and prognostic insights with the potential for advancement in clinical practice [86].

## 5. Limitations

While this review aims to provide a thorough understanding of the diagnostic and monitoring capabilities of PET/CT imaging in renal diseases, there may be some inherent limitations. First, the review may not have covered all of the relevant literature or research on the topic, potentially missing important findings or advancements in the field. Second, this review may have focused on a specific subset of renal diseases or PET/CT imaging techniques, overlooking other relevant aspects or applications. Additionally, this review may not have provided sufficient detail on specific PET/CT imaging radiotracers, or image interpretation methods.

## 6. Conclusions

In conclusion, this thorough review highlights the expanding applications of PET/CT in non-tumorous renal diseases. Its ability to provide comprehensive insights into renal structure and function across various conditions offers opportunities for early detection, precise diagnosis, and tailored treatment strategies. From AKI to CKD, PET/CT facilitates dynamic monitoring, treatment response assessment, and complication identification. PET/CT shows significant promise. PET/CT imaging helps in the diagnosis and evaluation of various renal disorders, making it helpful for timely intervention, particularly in AKI and drug-induced nephritis. It helps assess disease extent and guide biopsies, offering a non-invasive alternative to traditional methods. Additionally, PET/CT effectively monitors disease progression and response to treatment in chronic kidney disease and renal fibrosis, aiding in proactive treatment adjustments and early identification of complications like renal abscesses.

However, further research and educational initiatives are crucial for its seamless integration into clinical practice, allowing for the optimal utilization of this valuable diagnostic tool in diagnosing and managing non-tumorous renal diseases.

## Figures and Tables

**Figure 1 jcm-13-04632-f001:**
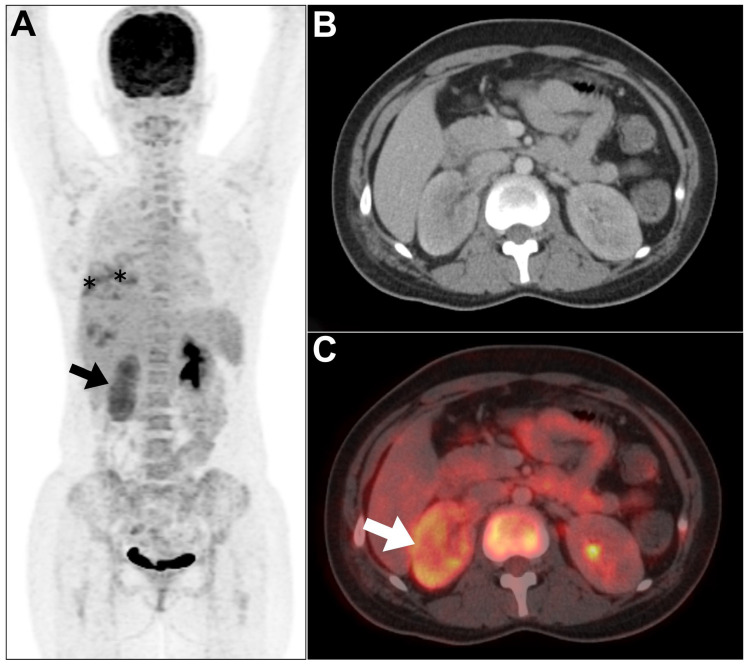
An example of acute pyelonephritis was encountered in a 35-year-old male patient who presented with fever and cough and was offered positron emission tomography/computed tomography (PET/CT) to investigate the source of pyrexia of unknown origin. (**A**–**C**) Maximum intensity projection (MIP), axial CT, and axial PET/CT images showed low-to-moderate-diffuse increased tracer uptake in the right renal cortex (arrows). The tracer activity within the left pelvicalyceal system is within physiological variation. Notably, there was a heterogenous increased tracer uptake in the right lung lower zone attributed to atelectasis (asterisks). This figure represents an unpublished case (original work of authors).

**Figure 2 jcm-13-04632-f002:**
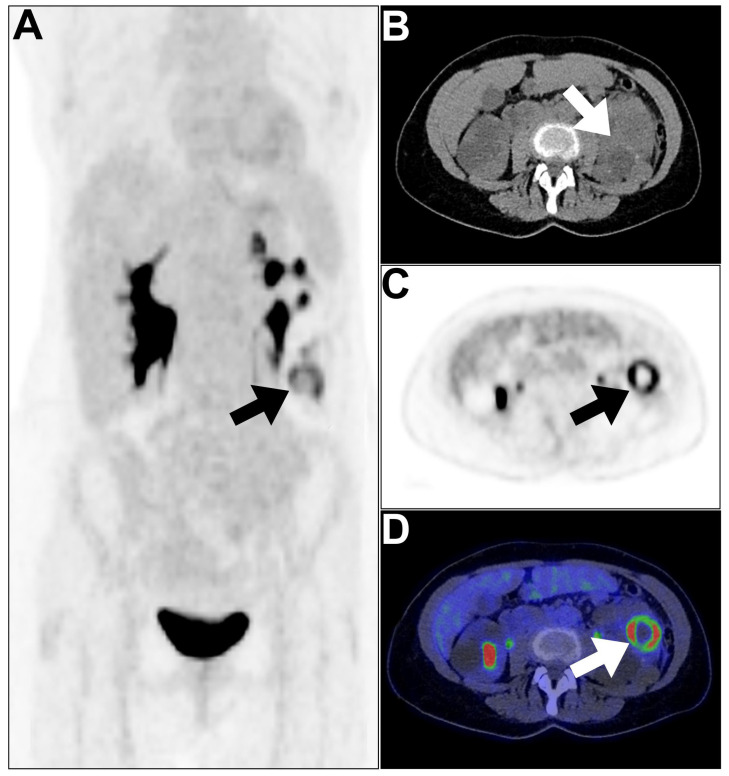
An example of an infected cyst in a patient with polycystic kidneys presented with persisting B symptoms for 8 weeks. (**A**–**D**) MIP, axial CT, axial PET, and axial PET/CT images revealed peripherally hypermetabolic focus surrounding the photopenic center in the inferior pole of the left kidney (arrows). The increased tracer uptake at the lower pole of the left kidney represents an infected cyst/abscess. This figure represents an unpublished case (original work of authors).

**Figure 3 jcm-13-04632-f003:**
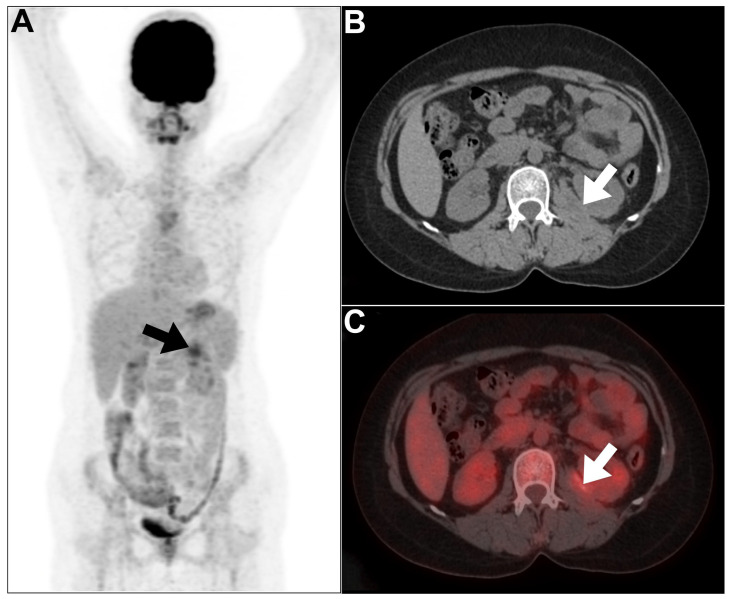
(**A**–**C**) MIP, axial CT, and axial PET/CT images of a 42-year-old male patient presented with hypermetabolic left perinephric soft tissue thickening (arrows) related to the condition of immunoglobulin G4 (IgG4-related nephropathy). This figure represents an unpublished case (original work of authors).

**Figure 4 jcm-13-04632-f004:**
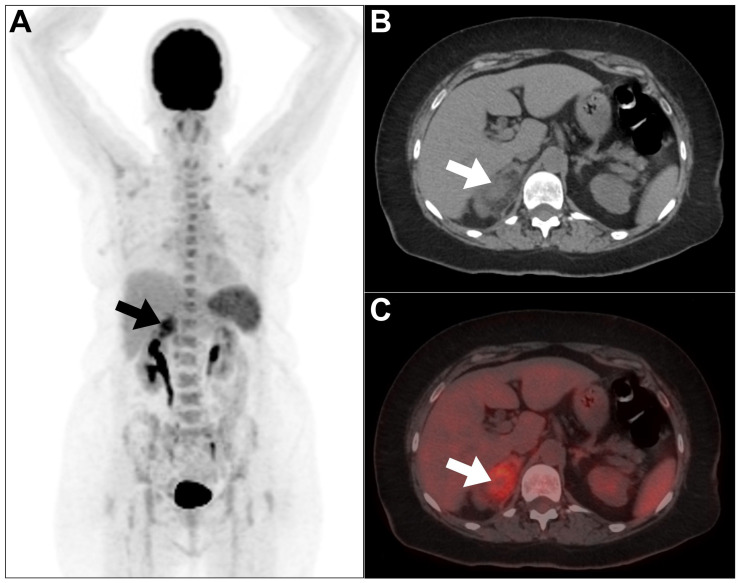
An example of a renal abscess encountered in a 39-year-old woman during an investigation of pyrexia etiology. (**A**) On the MIP image, there is a focal area of increased ^18^F-FDG uptake at the upper pole of the right kidney, corresponding to the site of abscess (arrow) seen on the axial CT image (**B**) with moderately increased tracer uptake (arrow) on the fused PET/CT image (**C**). Increased tracer uptake in the bone marrow and spleen can be best ascribed to an ongoing reactive process secondary to inflammation. This figure represents an unpublished case (original work of authors).

**Figure 5 jcm-13-04632-f005:**
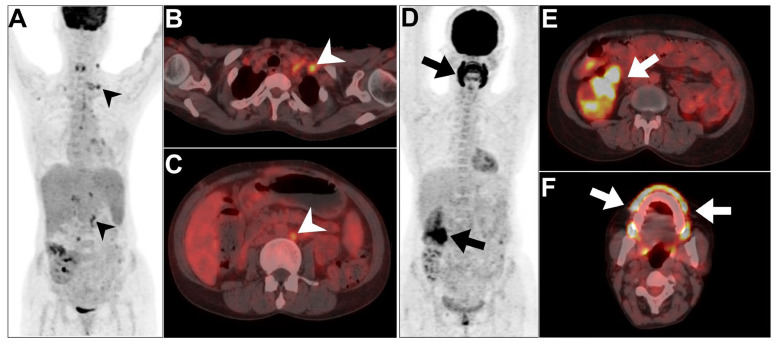
An example of two patients diagnosed with post-transplant lymphoproliferative disorder (PTLD). (**A**–**C**) MIP and axial views of PET/CT in a 45-year-old male patient revealed evidence of multiple hypermetabolic left supraclavicular and paraortic lymphadenopathies (arrowheads), compatible with PTLD diagnosis. (**D**–**F**) MIP and axial views of PET/CT in a 32-year-old male patient revealed evidence of diffusely hypermetabolic ascending colon deposition and hypermetabolic diffuse gingival localization indicative of an infective process. This figure represents an unpublished case (original work of authors).

**Figure 6 jcm-13-04632-f006:**
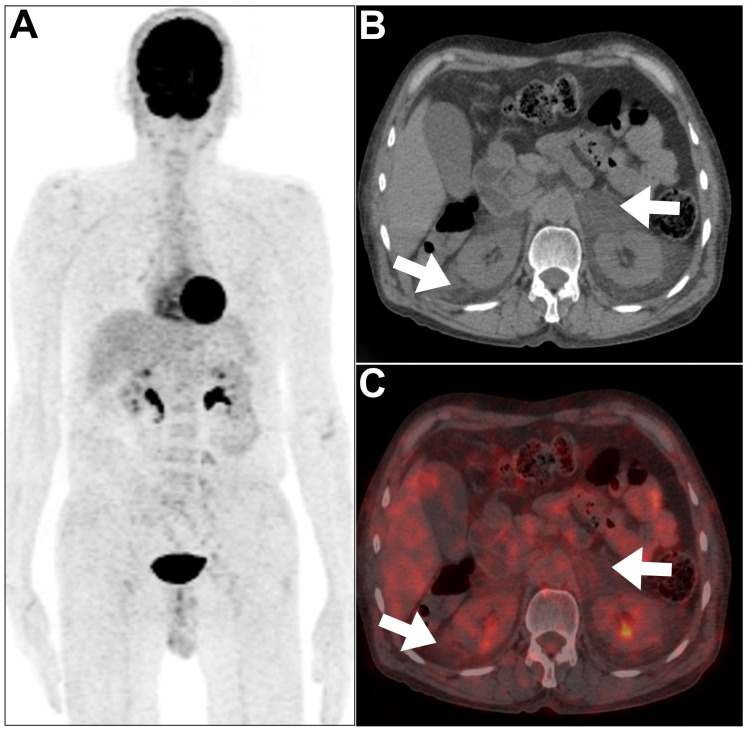
An example of renal amyloidosis was encountered in a 47-year-old male patient. (**A**–**C**) MIP, axial CT, and axial PET/CT revealed faint ^18^F-FDG uptake within bilateral perinephric soft tissue thickening (arrows). Biopsy confirmed amyloidosis. This figure represents an unpublished case (original work of authors).

**Table 1 jcm-13-04632-t001:** Optimal positron emission tomography (PET) tracers and imaging findings for non-tumorous renal diseases.

Renal Disease	PET Radiotracer	PET/CT Findings
Acute kidney injury	^18^F-FDG ^1^ and ^18^F-FAPI ^2^	Increased diffuse uptake, indicating inflammation and acute injury
Acute pyelonephritis	^18^F-FDG and ^68^Ga-FAPI	Focal areas of high uptake in the renal parenchyma, often with perirenal involvement
Autosomal Dominant Polycystic Kidney Disease	^18^F-FDG and ^18^F-WBC ^3^	Enhanced uptake in infected or inflamed cysts
Drug-induced nephritis	^18^F-FDG	Diffuse increased uptake throughout the renal parenchyma
Granulomatous nephritis	^18^F-FDG	Focal or diffuse uptake patterns corresponding to granulomatous inflammation
Autoimmune kidney diseases	^18^F-FDG and ^68^Ga-FAPI	Increased uptake in affected areas due to autoimmune-induced inflammation
Chronic kidney disease and renal fibrosis	^68^Ga-FAPI	Elevated uptake in fibrotic regions, correlating with the extent of fibrosis
Chronic pyelonephritis	^68^Ga-FAPI	Persistent, patchy increased uptake, indicating chronic inflammation
Renal transplantation	^18^F-FDG, ^15^O-H_2_O, and ^68^Ga-pentixafor	High uptake in cases of acute rejection; ^15^O-H_2_O shows perfusion deficits; ^68^Ga-pentixafor indicates inflammation and potential rejection
Renal histiocytosis	^18^F-FDG and ^68^Ga-FAPI	High uptake in histiocytic infiltrates within the kidneys
Renal amyloidosis	^18^F-FDG and ^123^I-SAP ^4^	Variable ^18^F-FDG uptake; ^123^I SAP shows high sensitivity and specificity for detecting amyloid deposits in renal tissue

^1^ FDG, fluorodeoxyglucose; ^2^ FAPI, fibroblast activation protein; ^3^ WBC, white blood cell; ^4^ SAP, serum amyloid protein.

## Data Availability

The data presented in this study are available upon request from the corresponding author. The data are not publicly available due to privacy.

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
