# Peer review of "A New Era for PET/CT: Applications in Non-Tumorous Renal Pathologies"

_jcm, 2024, doi:10.3390/jcm13164632_

Round 1

Reviewer 1 Report

Comments and Suggestions for Authors

This review manuscript has systemically summarized the role of PET/CT in the evaluation of non-tumorous renal diseases, which provided a comprehensive strategy of the non-invasive examination for the early detection, precise diagnosis, and dynamic monitoring of therapy  as well as complication identification of various kidney diseases. I have one point needed to address about the detection of Amyloidosis by scintigraphic imaging with radiolabelled SAP, it is suggested to supplement this part due to the important role in the assessment of systemic amyloidosis. 

Author Response

Comments 1: [ I have one point needed to address about the detection of Amyloidosis by scintigraphic imaging with radiolabelled SAP, it is suggested to supplement this part due to the important role in the assessment of systemic amyloidosis.] 

Response 1: Thank you for pointing this out. We agree with this comment. Therefore, we have added a paragraph discussing the role of radiolabelled SAP detection of renal Amyloidosis.  Added on page 15, second paragraph, lines 527-538 (highlighted in red). 

Reviewer 2 Report

Comments and Suggestions for Authors

The Review "A New Era for PET/CT: Applications in Non-Tumorous Renal Pathologies" presents a niche topic, particularly interesting for the medical world. Recommendations:

1.       The introduction needs more details about the importance of PET in the diagnosis of renal pathologies.

2.       Under the figures, specify their source.

3.       Add a discussion chapter in which the main similar studies in the literature on the effectiveness of PET/CT in the diagnosis of renal pathology and the results of these studies are presented.

4.       The conclusions need to be rewritten.

5.       Add a chapter on the limitations of the study.

Author Response

Comments 1: [The introduction needs more details about the importance of PET in the diagnosis of renal pathologies.] 

Response 1: Agree. We have, accordingly, revised this section to include a more detailed discussion on the importance of PET imaging in the diagnosis of various non-tumorous renal diseases. We believe these additions address the comment effectively and enhance the manuscript's clarity on the significance of PET imaging in renal diagnostics. Added on page 2, second paragraph, lines 55-67 (highlighted in red).

Comments 2: [ Under the figures, specify their source] 

Response 2: Thank you for pointing this out. All the figures represent an unpublished case (Original work of authors. Therefore, we have added this sentence under each figure.  Added on page 3, line 112 + page 6, line 181 + page 9, line 318 + page 12, line 424 + page 14, line 493 + page 16, line 542 (highlighted in red).

Comments 3: [Add a discussion chapter in which the main similar studies in the literature on the effectiveness of PET/CT in the diagnosis of renal pathology and the results of these studies are presented.] 

Response 3: Thank you for pointing this out.  We have, accordingly, added a discussion chapter to the manuscript. This new section presents an overview of the main studies in the literature regarding the effectiveness of PET/CT in diagnosing non-tumorous renal diseases. We believe this addition enhances the manuscript by providing a thorough examination of existing research and situating our work within the broader context of the field. Added on pages 16-18, lines 543-546 + 550-571 (highlighted in red).

Comments 4: [The conclusions need to be rewritten.] 

Response 4: Agree. We have, accordingly, revised the conclusion to better highlight PET/CT’s role in diagnosing and evaluating renal disorders, guiding timely intervention, and monitoring disease progression. Added on page 18, fourth paragraph, lines 603-609 (highlighted in red).

Comments 5: [Add a chapter on the limitations of the study.] 

Response 5: Thank you for pointing this out.  We have, accordingly, added a chapter on the limitations of the study. This new section addresses potential gaps in the literature, a focus on specific renal diseases or imaging techniques, and insufficient detail on certain PET/CT radiotracers and image interpretation methods. Added on page 18, third paragraph, lines 588-595 (highlighted in red).

Reviewer 3 Report

Comments and Suggestions for Authors

Dear editor: The manuscript addresses a topic of interest to nephrologists. New diagnostic and prognostic methods are useful in kidney diseases.

The work is a narrative review, not a systematic one, which decreases its scientific quality. In any case, the citations and writing are adequate.

The writing is appropriate and clear for the reader.

I suggest the authors make a table that summarizes: for each disease, the type of PECT that could be useful, as a guide. This could increase the scientific quality of the work.

Comments on the Quality of English Language

Not aaplicable

Author Response

Comments 1: [ I suggest the authors make a table that summarizes: for each disease, the type of PECT that could be useful, as a guide.] 

Response 1: Thank you for pointing this out. We agree with this comment. Therefore, have created a table summarizing the types of PET/CT imaging that could be useful for each renal disease discussed in the review. This table serves as a practical guide, providing a clear overview of the recommended PET/CT radiotracer for various conditions.  Added on pages 16-17, lines 547-549 (highlighted in red).

Round 2

Reviewer 2 Report

Comments and Suggestions for Authors

The authors have made the suggested modifications.